# Is Silence Golden? The Influence of Employee Silence on the Transactional Leadership and Job Satisfaction Relationship

**Yousef Ahmad Alarabiat ***  **and Serife Eyupoglu**

Business Administration Department, Faculty of Economics and Administrative Sciences, Near East University, Cyprus/Mersin 10, Nicosia 99010, Turkey
* Correspondence: yousefarabiat@gmail.com

**Abstract:** The literature has shown that a positive relationship between transactional leadership and job satisfaction in private and public organizations exist. This relationship is critical for sustained organizational performance; however, this relationship can be challenged by the existence of employee silence in the organizational setting. Based on self-determination theory, this study measured the impact of transactional leadership on job satisfaction as well as the part of employee silence in the leadership–satisfaction relationship in a public organizational setting. The study sample consisted of employees working at the Ministry of Justice in Jordon, and 450 questionnaires were applied with 357 useable questionnaires being returned. The data were tested through confirmatory factor analysis, correlation and regression analyses, and structural equation modeling. The results showed a strong positive relationship between transactional leadership and job satisfaction, though employee silence as a mediator indicated reduced job satisfaction. The importance of public organization leaders being more mindful towards the employee silence phenomenon and how it can be detrimental in the transactional leadership–job satisfaction relationship was highlighted.

**Keywords:** transactional leadership; job satisfaction; employee silence; public organizations; self-determination theory



## 1. Introduction

There is no doubt that sustainable organizational performance is a significant issue for private as well as public organizations. Workers who are more satisfied with aspects of their job, such as the work itself, remuneration, and their relationship with supervisors, tend to be more willing to reward the organization with favorable behaviors, including organizational performance [1]. This indicates that organizational performance stems from job satisfaction. In this sense, job satisfaction is actually considered as one of the significant employee outcomes and has been gaining noteworthy attention among practitioners and researchers for years [2].

The transactional leader encourages employees to meet expectations and in return rewards them for meeting predetermined performance standards. The literature has shown that a positive relationship between transactional leadership (TS) and job satisfaction does exist [3–5]. Therefore, the transactional leader is expected to generate greater levels of job satisfaction, and, in order for leaders to boost sustainable organizational performance, they need to ensure higher levels of employee satisfaction, which can been ensured through contingent rewarding. Transactional leadership is a style of leadership that has been researched and considered as a basis for successful leadership as well as a diversity of many other factors such as job satisfaction, employee performance, employee commitment, and organizational performance [2,6]. It is a style that leaders use to influence their followers/employees to reach a strong sense of commitment in order to direct them toward achieving the desired goals by adopting a rewards and punishment system. Transactional leadership can effectively cut down costs, increase organization production, monitor followers carefully so they stick to the rules, and achieve quick sustainable short-term goals [7].

In addition to that, it can guide the followers/employees to achieve the desired goals and at the same time brings satisfaction in the working environment, where such a high-level of satisfaction among the employees influences them to perform effectively in terms of keeping track of the organizational interests. Therefore, both the effectiveness and success of organizations are greatly influenced by this kind of leadership [8].

Even though there are many leadership styles, the transactional leadership style is regarded as one of the most common styles in organizations [9,10]. One reason for this may be because when an organization is in a stable condition (as most public organizations are), the transactional leadership style is often used and is very useful for the organization to cover balancing organization performance [11]. Mickson and Anlesinya [5] provide empirical support indicating that local government services utilizing transactional leadership behaviors, characterized by management-by exception and contingent rewards, are a better indicator of job satisfaction when compared to transformational leadership. Additionally, verbal rewards are included in the use of dependent rewards, one of the elements of transactional leadership. Managers in public enterprises often see verbal rewards as significant due to their limited access to monetary rewards and promotions.

Despite the importance of transactional leadership to the development of job satisfaction, empirical research has documented mixed results [12]. Numerous studies have discovered a positive and significant impact on employees' job satisfaction in organizations [13–15]. However, at the same time, other studies have reported a negative association [16–18]. Research to understand the mechanisms of positive and negative transactional leadership and how both of them can influence job satisfaction in different ways is required [19]. Additionally, according to previous research in leadership and the employee voice, there are several questions that remain unaddressed and need to be answered [20]. To bring employees' self-interest into line with organizational objectives, transactional leadership employs contingent rewards and/or sanctions. Employee self-interest is intended to motivate, steer, and sustain behavior toward those efforts and results when desired incentives (and undesirable sanctions/punishment) are dependent on particular efforts or results. Contingencies may be connected to monetary or near monetary benefits, such as bonuses, verbal rewards, such as praise, or consequences, such as firing [21]. Transactional leadership links the use of contingent rewards and sanctions to make individual employees seek their own self-interest while contributing to the achievement of organizational goals [22]. However, the use of sanctions/punishment may result in job dissatisfaction, eventually leading to employees feeling faulted and fearing that the sharing and offering of their opinions may lead to further consequences. This contributes to the development of what is known as "employee silence" [23].

Employee silence limits upward communication, keeps decision-makers in the organization in the dark about the organization's concerns, delays the making of timely choices, and lowers organizational performance, thus reducing employee satisfaction [24]. Therefore, for organizations, employee silence constitutes a great challenge with complicated concerns, as the effectiveness of leadership depends on the effectiveness of communication between leaders and employees [25]. Moreover, silence leads to low satisfaction and instability and harms both employees' performance and the organization's sustainable performance [26]. On the other hand, the employees' engagement and satisfaction with their organization or work are clearly shown in the employees' loyalty, which positively affects the performance of the organization when the employees are perceived as an essential resource in it. In other words, when employees have something important to say, they have to disclose it, and not be silent, to save the company from any future threat that may affect it [27].

In light of the above, the current study sought to better understand the impact of transactional leadership on job satisfaction as well as the role of employee silence in the leadership–satisfaction relationship. In this respect, this study attempts to investigate the relationship between transactional leadership and job satisfaction in public organizations in an effort to close the gaps in the literature. Leadership style and job satisfaction in public

organizations has received little research attention. Secondly, the study tests the effects of the different dimensions of transactional leadership that are contingent reward behavior, active management-by-exception behavior (active MBE), and passive management-by-exception behavior (passive MBE), on job satisfaction in order to understand positive and negative outcomes. Thirdly, understanding the devastating impact of employee silence on the relationship between transactional leadership and job satisfaction is urgently needed. Consequently, the study investigates the mediating effect of employee silence between transactional leadership and job satisfaction. Furthermore, the study will be carried out in a public organization in a developing country (Jordan). The few studies that have been conducted have been mainly in developed countries. Finally, this study attempts to understand the transactional leadership, job satisfaction, and employee silence relationship, drawing on the self-determination theory. Hence, the findings of this study should add to the body of knowledge on leadership in developing countries by highlighting the crucial role that transactional leadership plays in ensuring job satisfaction in public organizations and the detrimental effects of employee silence, an area that has received little attention in the literature.

## 2. Theoretical Framework and Development of Hypotheses

### 2.1. The Self Determination Theory

The self-determination theory (SDT) is an important concept that refers to the ability of individuals to make choices to reach their goals. This ability plays an important motivation role that allows people to feel they have control over their choices. SDT emphasizes there are three important basic needs that leaders should pay attention to (autonomy, competence, and relatedness). These demands are thought to be universal and essential for achieving everyone's needs for psychological health and individual's satisfaction [28,29]. Additionally, SDT emphasizes that trust is a crucial component of social exchange that promotes information sharing and increases the employees' voice in the organization [20]. While leaders always work on satisficing employee's needs, SDT suggests that some of the transactional leaders' behaviors can have negative impacts on their ability to fulfil the basic psychological needs of their followers and reduced sense of autonomy and competence. Through contractual obligations (rewards and punishment) transactional leadership focus on meeting standards [30,31]. In this study, transactional leadership and its connection with job satisfaction is introduced, as in prior research, and then how it can lead to employee silence as a result of external contingencies that can make it more challenging for transactional leaders to meet the fundamental psychological demands of their employees is discussed.

### 2.2. Transactional Leadership

Leadership styles play a significant role in influencing followers' performance, satisfaction, and organizational success and has been an important topic in the social sciences for many decades. A leader's style refers to the characteristics that the leader possesses (i.e., motivating, inspiring, and managing followers). Effective leadership is essential to achieving long-term sustainability for both individuals and organizations, and team success should be demonstrated by the leader in a trustworthy and reliable manner. Therefore, it is crucial to research different leadership styles and how they affect an organization and its followers. It is possible to also have successful leaders, but at the same time, they are despised by many employees because of the abusive style they use with their followers, and such a style may lead elite followers to leave the workplace, thus harming the organization. As there are many definitions of leadership, there are also numerous styles identified by researchers. From among these different leadership styles, researchers have defined some of the more distinctive styles, which are distinct in nature and need more focus in the studies in order to advance the right style of leadership to achieve the success of the organization in the current global competitive environment, and one of these distinct leadership styles is transactional leadership [1,32]

According to Bass [33], the majority of leadership research has conceptualized leadership as a transactional or cost-benefit exchange process. The theory of transactional leadership is based on the idea that interactions between leaders and followers form the basis of the relationships between them. The predominant belief is that the leader will be effective by making up for deficits through his or her behavior when the follower's environment is unable to provide the necessary direction and motivation. The transactional leader is straightforward with their followers regarding what is required of them and what they can expect in return [21]. In other words, the transactional leader encourages subordinates to meet expectations. According to Bass [33], there are three components for transactional leadership. Contingent reward, or dimension one of transactional leadership, is the practice of rewarding subordinates for meeting predetermined performance standards. In this respect, incentives are based on the level of performance and the amount of work put forward. Active and passive management-by-exception are the two styles of management-by-exception covered by dimensions two and three of transactional leadership. The leader only takes action through the management-by-exception technique when things go wrong and standards are not reached [34]. If performance targets are accomplished, the leader refrains from offering instructions and permits followers to carry on with their work as usual [35]. The active-type leadership is characterized by a leader who actively seeks out departures from the norm and responds when irregularities occur. Leaders who act only after deviations and anomalies have occurred are characterized by the passive form. Therefore, the distinction between the two is that the leader in the active dimension looks for deviations, whereas the leader in the passive dimension waits for issues to arise [11,35].

### 2.3. Transactional Leadership and Job Satisfaction

The extent of employees' satisfaction with their job has been an engaging topic for several years due to the rapid development of globalization and complexities and challenges in organizations and the environment, which have created interest for psychologists, academics, and administrators. The key to employee's comfort is their job satisfaction, and it shows in how well they perform at the workplace. According to Locke [36], job satisfaction is a pleasant or positive emotional state resulting from an assessment of one's work experience. It is a definition that is frequently used in studies on job satisfaction. One of the main reasons that contribute significantly to increasing job satisfaction among an organization's members is the great influence the leader has on followers and the way they work. Ineffective management styles are one of the fundamental factors for the low levels of job satisfaction in organizations [37]. In order to raise employee work satisfaction levels, the leaders have to develop a solid trust with employees through empowering them to take part in decision-making, establish open communication lines, give them more responsibility and independence, and construct a rewards system [38]. Transactional leaders often direct and push forward more effectively by clarifying followers' tasks and role demands by linking such roles and demands with rewards and punishments. Thus, transactional leaders can make their followers happy by rewarding their dedication to work as well as punishing defaulters [39,40]. Accordingly, extrinsic motivators, such as contingent rewards and punishments, which are typical characteristics of transactional leadership, can shape employees' levels of satisfaction (and dissatisfaction) towards their work. The study on the influence of leadership style on the job satisfaction of non-governmental organization (NGO) employees and the mediating role of psychological empowerment confirmed the previous results of studies in terms of the direct positive relationship between transactional leadership and employee job satisfaction [41]. The success of transactional leadership lies in achieving satisfaction between the leader and the followers, as it depends on the performance appraisal system. However, this type of leadership focuses on two aspects: the reward system and punishments. Through the reward system, the followers can become satisfied but meet the minimum expectations to avoid punishment. In this type of leadership, the leader can negotiate with followers regarding the results he/she seeks to achieve [42]. In a study on the effect of transactional leadership as a type of leadership

on employee performance, the results showed a high correlation value for transactional leadership with employee performance, as the leadership style contributes significantly to motivation and job satisfaction when individuals have a sense of emotional attachment with their organization [43]. While the effect of transactional leadership on job satisfaction has been excellent and fair in many previous studies, in studying the effect of transactional leadership on job satisfaction in the selected retail outlets, the findings indicate a very low effect of transactional leadership on job satisfaction [44]. According to Nazim and Mahmood's [19] study, all dimensions of transactional leadership as one of the leadership styles were measured, and the results showed that all dimensions (contingent reward, active management-by-exception, and passive management-by-exception) had a different impact on job satisfaction, where the greatest influence was the effects of active management-by-exception and the lowest influence was found for passive management-by-exception. Dartey-Baah and Ampofo [15] researched how transactional leadership styles influenced job satisfaction in the manufacturing industry; their findings concluded that all dimensions of transactional leadership have a significant impact, especially for the contingent reward and active management-by-exception dimensions.

Hence, this study hypothesizes that:

**Hypothesis 1 (H1).** *The contingent reward dimension of transactional leadership has a positive impact on job satisfaction.*

**Hypothesis 2 (H2).** *The active MBE dimension of transactional leadership has a positive impact on job satisfaction.*

**Hypothesis 3 (H3).** *The passive MBE dimension of transactional leadership has a positive impact on job satisfaction.*

*2.4. Transactional Leadership and Employee Silence*

Employees are generally regarded as an organization's most significant resource. They have essential roles in innovation, transformation, and creativity—all central in accomplishing organizational goals—yet they typically opt to keep quiet rather than share their valuable opinions and worries about issues at work. Managers need to understand why their employees behave the way they do. If organizations intend to achieve their performance objectives, employee silence must generally be understood. Employees can offer important ideas and thoughts, information, and suggestions on how to improve the workplace. Even though employees might occasionally express themselves, they might also remain silent and repress their ideas and thoughts at other times.

In the past, concepts of silence assumed that nothing bad could happen because there was no emergence of controversy; thus, the primary definitions of silence were equated with loyalty. However, researchers have shown that employee silence can lead to invalidation of the desired results and failure to achieve goals at the required level [45,46]. Silence as the absence of voice in organizations means the absence of one of the communication forms: argument, objection, awareness, motivation, and support. Previous research also showed that employee silence could embrace various causes according to the fundamental motivations that make individuals silent, so researchers recognized the importance of constantly and widely examining the phenomenon of silence in organizations [47]. Morrison and Milliken [48] suggested that when employees prefer to remain silent about organizational matters in the workplace, this leads to silence becoming a collective behavior known as "organizational silence". In fact, employee silence represents an inefficient organizational process that leads to a waste of costs and efforts and an absence in the level of collective participation in developing plans, strategies, and providing solutions [49]. Therefore, transactional leaders must pay attention to employees who withhold information that may be useful to the workplace or organizations, whether intentionally or not, and try to motivate them, raise their voices, and disturb the silence to prevent any threats to the organization.

Therefore, the strong and effective leadership behavior practiced by the management can play an essential role in inspiring employees and exchanging information among team members, thus avoiding organizational silence and increasing levels of satisfaction [50,51].

Transactional leadership can have a negative impact on employees' behaviors. This negative impact can be detected by the logic of SDT. Whereas the SDT supports the fundamental requirements of an individual, including autonomy, competence, and relatedness, SDT emphasizes the fulfillment of these needs to reach for an outstanding level of follower satisfaction, but thwarting of these needs leads to negative results, thereby lower satisfaction. According to SDT, unique transactional leadership behaviors may either facilitate or obstruct the fulfilling of satisfaction requirements through strong influence over these people's fundamental needs [28]. Among the transactional leadership behaviors, active management-by-exemption involves an exceptional level of monitoring of the follower's performance and taking active corrective procedures when errors are seen [42]. As suggested by SDT, the active monitoring, pressure, and threat of receiving the punishment that leaders practice on their followers may lead to a reduced sense of incompetence and autonomy. Hence, the feeling of job dissatisfaction and employees' preference to commit silence [28]. However, active management-by-exception may decrease levels of autonomy and competence (i.e., job satisfaction) to a greater extent than passive management-by-exception because passive management-by-exception behavior gives freedom to followers to behave as they choose. Moreover, because SDT predicts that followers may focus more on gaining short-term gains from the rewarded behaviors than on long-term rewards, it is likely that contingent rewards will decrease work satisfaction [10,52]. Transactional leaders emphasize reward and punishment behavior to accomplish organizational goals or required performance. Cognitive evaluation theory, a sub theory of SDT, suggests that reward and punishment behavior is deeply tied to performance, which may reduce the sense of autonomy and competence for employees, so perhaps transactional leadership can reduce intrinsic motivation for employees and increase silence by reducing the sense of autonomy and competence for employees [6,28,29].

The constant competition between organizations and the technological developments that organizations are witnessing today make leadership styles and the need to avoid silence in organizations the focus of attention for practitioners. Employees who perceive greater support from their organizations can participate more in the expected results of the organization by sharing their knowledge and offering solutions, and they can speak up, even if they think differently than others. On the other hand, the perceived organizational support can be compromised by the silence of the organization; employees who take the side of silence in the organization may then feel less support from the organization, and vice versa. When an employee has appropriate support and leadership in the workplace, it helps them raise their voice and motivates them. The consequences of individuals' silence have not been widely studied [52].

Hence, this study hypothesizes that:

**Hypothesis 4 (H4).** *The contingent reward dimension of transactional leadership has a negative impact on employee silence.*

**Hypothesis 5 (H5).** *The active MBE dimension of transactional leadership has a positive impact on employee silence.*

**Hypothesis 6 (H6).** *The passive MBE dimension of transactional leadership has a positive impact on employee silence.*

### 2.5. Employee Silence and Job Satisfaction

Employee silence is considered as one of the harmful factors in the work environment and organizations; it leads to a decreased level of job satisfaction, increased absenteeism, increased employee turnover, and many harmful behaviors. Communication is considered

the key to avoiding silence in the work environment. If the employee remains silent, this affects the communication between members of the organization [53] and contributes to many damaging outcomes in organizations, such as killing innovation and creating inadequate strategic planning, as well as leading to massive financial losses to companies. Over time, employee silence creates employees who are indifferent to the nature of work and to the commands of the leader [54]. Unfortunately, when companies and organizations face considerable financial losses, leaders or managers tend to try to compensate for these losses, ignoring the fact that their employees become indifferent because of their silence, as they do not take the initiative to solve problems as they arise [26]. According to a study by Demirtas [53], which researched the link between organizational values, job satisfaction, organizational silence, and commitment, silence is one of the essential factors that negatively influences employee job satisfaction.

Hence, this study hypothesizes that:

**Hypothesis 7 (H7).** *Employee silence is negatively related to job satisfaction.*

*2.6. The Mediating Role of Employee Silence in the Relationship between Transactional Leadership and Job Satisfaction*

According to Morrison and Milliken [48], silence can lead to feelings among employees that they are not valued or in charge of their work, both of which will decrease internal motivation, engagement, and pleasure at work. Employee silence, according to empirical study, diminishes job satisfaction and leads to more job burnout [55,56]. According to the SDT view, it is suggested that transactional leadership may reduce the perceived autonomy and competence of employees because it emphasizes the principles of reward and control, which can demotivate the level of satisfaction between individuals and increase silence [24,28,57]. Therefore, the current study contends that the relationship between transactional leadership behaviors and job satisfaction will be significantly impacted by employee silence.

Hence, this study hypothesizes that:

**Hypothesis 8 (H8).** *Employee silence mediates the relationship between the contingent reward dimension of transactional leadership and job satisfaction.*

**Hypothesis 9 (H9).** *Employee silence mediates the relationship between the active MBE dimension of transactional leadership and job satisfaction.*

**Hypothesis 10 (H10).** *Employee silence mediates the relationship between the passive MBE dimension of transactional leadership and job satisfaction.*

In light of the above discussion of the study variables, the research model is presented in Figure 1 below:

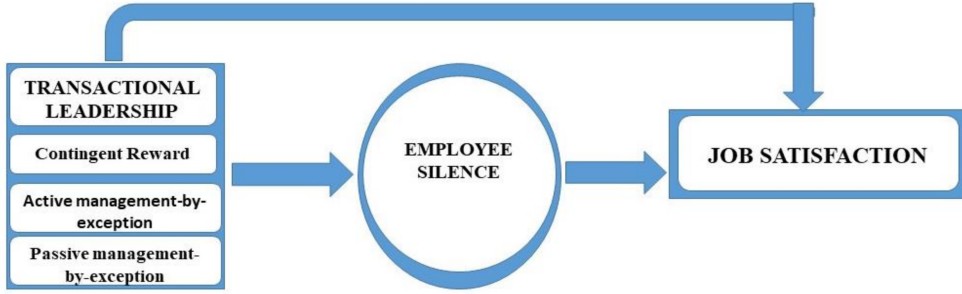

**Figure 1.** Research Model.

## 3. Methodology

### 3.1. Research Population and Sample

This cross-sectional quantitative study used convenience sampling, together with a survey method to collect data from employees working at the Ministry of Justice in Jordan. A sample size of 450 employees was seen as appropriate because there were 4000 employees overall at the time of the study. Through the use of convenience sampling, the questionnaire was distributed to 450 employees, with 357 useable questionnaires being returned (80% response rate) and was considered suitable for research and met the sample size conditions [58,59]. Employee respondents were given the questionnaire in either print copy or electronic form. The importance and goal of the study were explained to the respondents, and they received assurances that their rights would be protected. The respondents were also told that participation was optional and that the information would only be utilized for academic study by the authors. The questionnaire was given to respondents who would rather complete it on paper, and they were instructed to fill it out and send it back to the authors. Those who selected the electronic version received a link via email. Respondents were not given a deadline for completing and submitting the questionnaire. Between September and December of 2021, the data were collected.

### 3.2. Measures

There were two main parts for the questionnaire; part one included demographic and occupational information for respondents, and the second part consisted of three sections: The Transactional Leadership (TS) Scale, The Employee Silence Scale, and The Job Satisfaction Scale. Transactional leadership practices were measured using the measurement scale adopted from Akhigbe et al. [60]. The scale includes 13 items that measured the three sub-dimensions of transactional leadership: contingent reward, active management-by-exception, and passive management-by-exception. Respondents were asked to rate their perceptions of transactional leadership on a five-point Likert scale extending from 1 (strongly dissatisfied) to 5 (strongly satisfied). The following is a sample item: "My supervisor makes clear what one can expect to receive when performance goals are achieved".

The Employee Silence Scale consisted of five items and was adopted from Dong & Chung [52]. The scale consisted of five items, and again, on a five-point Likert scale, the respondents were asked to rate their level of satisfaction with each of the items. The following is a sample item: "I remain silent when I have information that might help prevent an incident in my workgroup".

The Job Satisfaction Scale was adopted from Bond [61]. The scale consisted of nine items and respondents were asked to show their level of satisfaction for each of the items on a five-point Likert scale. The following is a sample item: "I feel happy when I am working intensely".

### 3.3. Data Analysis

The suggested model was estimated using covariance-based structural equation modeling (SME) through AMOS. This approach to SEM is suitable for confirmatory research; it is flexible because it uses simple drawing tools and a compelling data analysis method, especially with reflective measurement [62]. It also estimates both the measurement model and the suggested relationships in one model and allows error control while testing the suggested relationships [63,64]. Data analysis using this approach requires the estimation of both the measurement model and the structural model to validate the tenability of the suggested model [65,66].

## 4. Results

### 4.1. Descriptive Statistics of Respondents

Table 1 summarizes the demographic and occupational characteristics of respondents. Just over half of the respondents (53.8%) were female, and 53.2% of respondents held a bachelor's degree. Most of the respondents (84.6%) were married. With regard to work

experience in the organization, the majority of the respondents (25.5%) had between 10 to 14 years of experience. As for experience in the public sector, the majority of the respondents, 30%, had between 10 to 14 years of experience.

**Table 1.** Demographic characteristics of respondents.

| Demographic | Characteristics | Frequency | Percent |
|---|---|---|---|
| Gender | Female | 192 | 53.8 |
| | Male | 165 | 46.2 |
| Age | Between 30 and 39 years | 153 | 42.9 |
| | Between 40 and 50 years | 151 | 42.3 |
| | Less than 30 years | 20 | 5.6 |
| | More than 50 years | 33 | 9.2 |
| Status | Married | 302 | 84.6 |
| | Unmarried | 55 | 15.4 |
| Education | Bachelor | 190 | 53.2 |
| | Diploma | 78 | 21.8 |
| | Postgraduate | 89 | 25.0 |
| Experience | Less than 1 year | 9 | 2.5 |
| | Between 1 and 3 years | 28 | 7.8 |
| | Between 4 and 6 years | 44 | 12.3 |
| | Between 7 and 9 years | 27 | 7.6 |
| | Between 10 and 14 years | 91 | 25.5 |
| | Between 15 and 20 years | 89 | 24.9 |
| | More than 20 years | 69 | 19.3 |
| Experience in public sector | Less than 1 year | 13 | 3.6 |
| | Between 1 and 3 years | 19 | 5.3 |
| | Between 4 and 6 years | 26 | 7.3 |
| | Between 7 and 9 years | 35 | 9.8 |
| | Between 10 and 14 years | 108 | 30.3 |
| | Between 15 and 20 years | 88 | 24.6 |
| | More than 20 years | 68 | 19.0 |
| | Total | 357 | 100.0 |

*4.2. Measurement Model Assessment*

The suggested measurement model comprises five latent variables, which were created as reflective measures according to prior research. Confirmatory factor analysis (CFA) was conducted to examine the reliability, convergent validity, and discriminant validity of the latent variables. As commended by methodologists, composite reliability (CR) and Cronbach's alpha are adequate for reliability assessment; the coefficient of each should be greater than 0.7 for satisfactory reliability [67,68]. The extracted average variance (AVE) provides significant insight to assess convergent validity; each construct should demonstrate an AVE of 0.5 or above for sufficient convergent validity [69]. Item loadings also offer further evidence of convergent validity; items should load largely ($\geq$ 0.5) on their postulated constructs. The Heterotrait–Monotrait (HTMT) ratio has recently been introduced as a robust measure of discriminant validity [70,71]. The HTMT value between a pair of constructs should typically be below 0.85 for acceptable discriminant validity.

A satisfactory fit to the observed data should be shown by the proposed measurement model. There are numerous goodness-of-fit indicators to assess a model. The most frequently used indicators include the chi-square ($\chi^2$), the ratio of $\chi^2$ to the degree of freedom ($\chi^2$/df), the comparative fit index (CFI), the standardized root mean square residual (SRMR), and the root mean square error of approximation (RMSEA). The recommended cutoff values for excellent model fit suggest that $\chi^2$/df should be small (<3), CFI > 0.95, RMSEA < 0.08, SRMR < 0.06 [72]. CFA was conducted on our measurement model. One item (CME4) was unreliable due to low loading (0.3). This item was removed from the

model. Table 2 shows the assessment of reliability and convergent validity after the second round of CFA. The results indicate that the coefficients of composite reliability and Cronbach's alpha are in acceptable ranges (>0.7). They range between 0.764 for active management-by-exception and 0.892 for contingent reward. The values of AVE also satisfy the recommended cutoff criteria (>0.5) as they range between 0.633 for contingent reward and 0.588 for active management-by-exception. Item loading ranges were between 0.919 and 0.550, which is well above the recommended cutoff (0.5). All of the above figures confirm that the constructed measures were convergently valid and reliable. Table 3 shows the estimation of HTMT ratios between the constructs. The HTMT ratios range from 0.060 to 0.491, which signifies that the constructs are independent of each other, and hence, discriminant validity is evident.

**Table 2.** Convergent validity and reliability.

| Construct Name | Cronbach's Alpha ($\alpha$) | CR | AVE | Item Code | Loading |
|---|---|---|---|---|---|
| Contingent reward | 0.899 | 0.896 | 0.633 | CR1 | 0.770 |
| | | | | CR2 | 0.877 |
| | | | | CR3 | 0.748 |
| | | | | CR4 | 0.732 |
| | | | | CR5 | 0.842 |
| Employee silence | 0.892 | 0.883 | 0.601 | ES1 | 0.734 |
| | | | | ES2 | 0.826 |
| | | | | ES3 | 0.817 |
| | | | | ES4 | 0.767 |
| | | | | ES5 | 0.728 |
| Job satisfaction | 0.950 | 0.941 | 0.642 | JS1 | 0.665 |
| | | | | JS2 | 0.845 |
| | | | | JS3 | 0.853 |
| | | | | JS4 | 0.837 |
| | | | | JS5 | 0.919 |
| | | | | JS6 | 0.78 |
| | | | | JS7 | 0.753 |
| | | | | JS8 | 0.738 |
| | | | | JS9 | 0.793 |
| Active management-by-exception | 0.777 | 0.764 | 0.588 | CME1 | 0.810 |
| | | | | CME2 | 0.897 |
| | | | | CME3 | 0.550 |
| Passive management-by-exception | 0.889 | 0.871 | 0.629 | PME1 | 0.853 |
| | | | | PME2 | 0.871 |
| | | | | PME3 | 0.711 |
| | | | | PME4 | 0.725 |

**Table 3.** Ratios of Heterotrait–Monotrait (HTMT).

| Construct Name | 1 | 2 | 3 | 4 | 5 |
|---|---|---|---|---|---|
| Contingent reward (1) | | | | | |
| Employee silence (2) | 0.102 | | | | |
| Job satisfaction (3) | 0.356 | 0.140 | | | |
| Active management-by-exception (4) | 0.489 | 0.250 | 0.212 | | |
| Passive management-by-exception (5) | 0.339 | 0.491 | 0.054 | 0.060 | |

The proposed measurement model seems to be a good fit for the data ($\chi^2$ = 532.118; df = 279; $\chi^2/\mathrm{df}$ = 1.907; CFI = 0.962; RMSEA = 0.050; SRMR = 0.050).

A measurement model can be threatened by the problem of common method bias (CBM), which happens when the instrument introduces a bias by causing variations in

responses [73]. One of the most common ways to examine CMB is to constrain the measurement model by a common latent factor (CLF). The CLF reflects the common variation in a measurement model [74]. The model fit should be compared between the constrained and unconstrained models. Significant differences between the two models indicate the issue of CMB. This study estimates the measurement model with a CFL. The comparison between the constraint and unconstraint models indicates that there are insignificant differences between them ($\Delta\chi^2$/df = 0.158; $\Delta$CFI = 0.009). Therefore, CMB would not bias the inferences of this study.

### 4.3. Structural Model Assessment

Structural model assessment includes an examination of the coefficients of proposed associations and whether the proposed model is a good fit for the observed data. Figure 2 depicts the structural model of this study and shows the estimation of the hypothesized relationships.

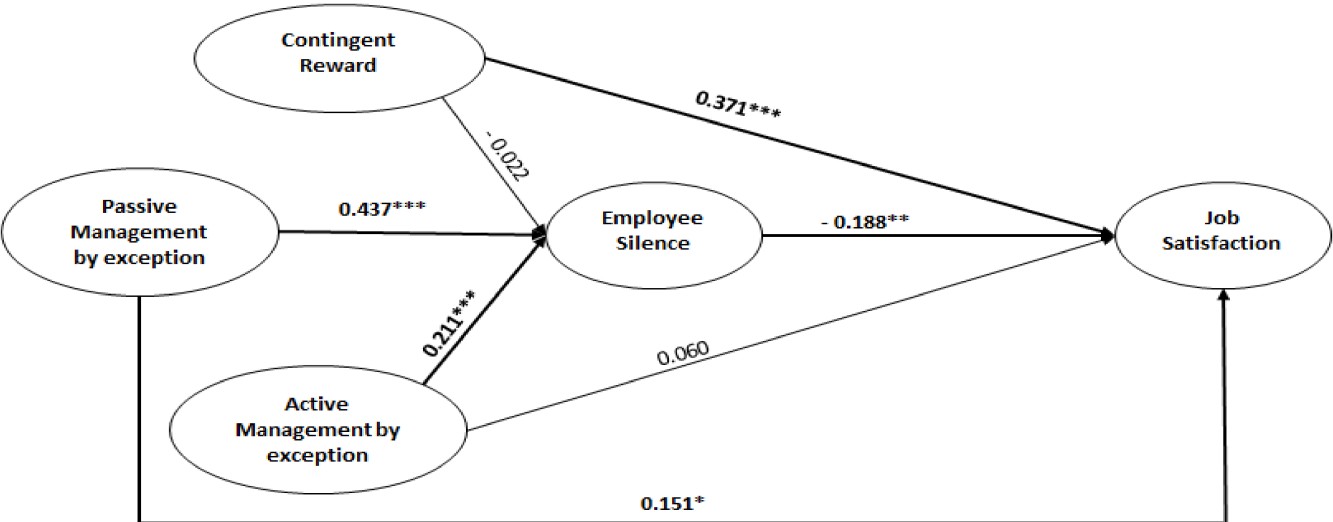

**Figure 2.** The structural model. Significance of Estimates: *** $p < 0.001$; ** $p < 0.010$; * $p < 0.050$.

The full estimation result is presented in Table 4. The model explains approximately 15.9% and 28.4% of the variance in job satisfaction and employee silence. The proposed measurement model seems to be a good fit for the data ($\chi^2$ = 532.118; df = 279; $\chi^2$/df = 1.907; CFI = 0.962; RMSEA = 0.050; SRMR = 0.050). The results show that contingent reward (path coefficient = 0.371, $p < 0.001$) and passive management by exception (path coefficient = 0.151, $p < 0.01$) contribute positively and significantly to job satisfaction. Therefore, there is strong evidence that support hypotheses H1 and H3. The results also indicate that active management by exception does not have a significant direct effect on job satisfaction (path coefficient = 0.060, $p > 0.05$). Accordingly, there is no empirical support for hypothesis H2, and it is rejected. The results indicate that employee silence is positively and significantly affected by passive management by exception (path coefficient = 0.437, $p < 0.01$) and active management by exception (path coefficient = 0.211, $p < 0.01$), but not by contingent reward (path coefficient = −0.022, $p > 0.05$). Accordingly, hypotheses H5 and H6 are supported, and hypothesis H4 is rejected. Moreover, employee silence does significantly reduce job satisfaction (path coefficient = −0.188, $p < 0.01$), providing support for hypothesis H7.

**Table 4.** Estimation of the structural model.

| Hypothesized Association | Standardized Estimate | S.E. | t-Value | *p*-Value |
|---|---|---|---|---|
| H1: Contingent reward → Job satisfaction | 0.371 | 0.063 | 5.386 | 0.000 |
| H2: Active management by exception → Job satisfaction | 0.060 | 0.059 | 0.940 | 0.347 |
| H3: Passive management by exception → Job satisfaction | 0.151 | 0.056 | 2.078 | 0.038 |
| H4: Contingent reward → Employee silence | −0.022 | 0.065 | −0.363 | 0.716 |
| H5: Active management by exception → Employee silence | 0.211 | 0.065 | 3.392 | 0.000 |
| H6: Passive management by exception → Employee silence | 0.437 | 0.06 | 6.472 | 0.000 |
| H7: Employee silence → Job satisfaction | −0.188 | 0.061 | −2.723 | 0.006 |

To examine whether organizational silence suppresses (a negative confounding mediator) the positive impact of transactional leadership dimensions on job satisfaction, the mediation analysis procedures suggested by Preacher and Hayes [75] was followed. According to [75], a mediation effect is evident when the indirect effect between an independent variable and a dependent variable is significant. The significance of the indirect effect should be determined using a bootstrapping test that relies on confidence intervals. To offer empirical evidence of a mediation effect, the confidence intervals should not include zero. Table 5 shows the estimation of the indirect effects between transactional leadership dimensions and job satisfaction through employee silence. The results indicate that employee silence does suppress the positive impact of passive management-by-exception (Indirect path coefficient = −0.082, CI = −0.120; −0.023; $p < 0.01$) and active management-by-exception (Indirect path coefficient = −0.040, CI = −0.082; −0.011; $p < 0.05$), but not the positive impact of contingent reward (Indirect path coefficient = −0.004, CI = −0.018; 0.030; $p > 0.05$). Accordingly, hypotheses H9 and H10 are supported, and hypothesis H8 is rejected.

**Table 5.** Estimation of the indirect effects.

| Indirect Path | Standardized Estimate | Lower CI | Upper CI | *p*-Value |
|---|---|---|---|---|
| H8: Contingent reward → Employee silence → Job satisfaction | 0.004 | −0.018 | 0.030 | 0.646 |
| H9: Active management by exception → Employee silence → Job satisfaction | −0.040 | −0.082 | −0.011 | 0.011 |
| H10: Passive management by exception → Employee silence → Job satisfaction | −0.082 | −0.120 | −0.023 | 0.009 |

## 5. Discussion

Despite the frequent use of transactional leadership (TS) in government organizations, the existing literature has yet to determine how transactional leadership (as defined by contingent rewards, active management-by-exception, and passive management-by-exception) is related to job satisfaction through the mediating role of employee silence. More in-depth studies need to be conducted because the literature on employee silence is lacking.

This study measured the impact of transactional leadership on job satisfaction as well as the role of employee silence in the leadership–satisfaction relationship in a public organizational setting. Drawing on the self-determination theory (SDT), the study attempted to address these aforementioned points.

The study model demonstrated a direct relationship between transactional leadership and job satisfaction. This result is in line with previous studies. For instance, a study conducted amongst federal government employees indicated that transactional leadership positively influenced job satisfaction [6]. Likewise, in a study conducted amongst full time faculty members, transactional leadership was also found to have a positive influence on (outer) job satisfaction [32]. Similarly, a study conducted amongst NGO employees

found that a positive direct relationship existed between transactional leadership and job satisfaction [41].

The study model also demonstrated that transactional leadership behaviors predicted employee silence and that it also adversely impacted job satisfaction at work via an indirect effect. The literature indicates mixed findings in regards to the transactional leadership and employee silence relationship. In a study conducted amongst project team members, transactional leadership was found to be significantly negatively related to the defensive and prosocial dimensions of employee silence but not to the acquiescence dimension of employee silence [76]. Another study conducted amongst employee working in medium-sized enterprises reported that the acquiescence and defensive dimensions of employee silence increased with transactional leadership practices [77].

Furthermore, the study's findings additionally revealed that in the public organizational setting, the positive impact of transactional leadership on job satisfaction is suppressed by employee silence, as the transactional leadership behaviors active management-by-exception and passive management-by-exception have a significant positive impact on employee silence, though contingent reward behavior did not impact employee silence.

Finally, the study results indicated that an employee's silence serves as a mediator in the proposed relationship, where there are negative effects through the indirect relations and positive effects for direct relations on job satisfaction. In light of the above, it is hoped that this study contributes to the understanding of a number of important theoretical and practical criteria for establishing effective transactional leadership within the workplace.

## 6. Theoretical and Practical Implications Section

### 6.1. Theoretical Implications

The association between transactional leadership and job satisfaction has been largely investigated in prior research. The empirical findings reveal positive and negative associations between them, with limited explanation as to why transactional leadership can have negative implications to job satisfaction. Grounding on the SDT, this study was an attempt to describe the mechanism through which transactional leadership behaviors can negatively affect job satisfaction. Although transactional leadership involves a rewards and control approach if the tasks are not done, our study proposes that these practices could lead to employee silence and motivate employee silence and demotivate employees by lowering their perceived independence and ability [78]. Leaders adopting transactional leadership may negatively use reward practices through withholding bonuses, or not allowing for time off, thus contributing to the development of employee silence in the organization, where employees feel faulted and have a fear of sharing and offering their opinions for fear of the consequences. Accordingly, transactional leadership practices can contribute to the appearance of what is known as "employee silence". This idea is new and innovative in demonstrating the interplay between transactional leadership and job satisfaction. This study shows that employee silence can be a suppressor of the positive impact of transactional leadership. Moreover, this study examined whether the transactional leadership practices/dimensions have different impacts on job satisfaction and employee silence, and the study shows that contingent reward does not impact employee silence; however passive and active management-by-exception increases employee silence. Passive management-by-exception is related to leader passivity, which means that employee autonomy is relatively uninhibited because these leaders keep their followers at a distance, and active management-by-exception is associated with leaders who exhibit this behavior and are more active in monitoring their followers. Our findings confirm that transactional leadership behavior (with its dimensions) can have both positive and negative consequences for followers. Indeed, the two transactional leadership dimensions, active management-by-exception and passive management-by-exception, have a negative indirect effect on job satisfaction via employee silence (i.e., competence and autonomy). However, the negative indirect effect of employee silence on job satisfaction is offset by the positive direct effect of active management-by-exception and passive management-by-exception.

As a result, consistent with Thomas' [79] findings, when investigating and analyzing mediating variables, adding the mediator to the mediational chain may result in both positive and negative indirect effects.

*6.2. Practical Implications*

As the modern organizational environment becomes more complex, organizations need to break the silence by encouraging employees to speak up and share their thoughts and opinions. Employees avoid speaking up and prefer to remain silent in organizations where leaders do not take into consideration their attitudes, concerns, and remarks, which discourages them from trying to address organizational issues [80]. The existence or non-existence of employee silence is significantly influenced by leadership styles, so understanding the positive and negative influences of leadership styles on employee silence is crucial for sustainable leaders. Additionally, there is a link between employee silence behavior and employee job satisfaction. Leaders have to encouraging employees to contribute their opinions and ideas. By doing so, leaders are in fact aiding in the development of trust between themselves and their employees. This in turn will contribute to organizations' ability to deal with unforeseen situations and aid in avoiding any deviations [20].

The findings of this study have significant implications for practice, especially for the public organizational setting for which limited literature exists. In this sense, the results of this study can be used as a guide to assist public organizations in developing programs to educate their managers/leaders to recognize the influences of transactional leadership and understand how this style of leadership can be used to minimize employee silence rather than promote it. Likewise, this will also result in higher levels of job satisfaction amongst employees, which will in turn, in the long run, influence sustainable organizational performance.

The existence of employee silence in any organization indicates that employees are not provided with the opportunity to raise their voices. Thus, it is necessary for organizations, public as well as private, to create environments where employees feel free to express and share their ideas, creative solutions, and thought, and be able to interact with their leaders. One way to accomplish this is through improving the social exchange relationship between leaders and employees. Transactional leadership is concerned with a dynamic exchange between leaders and their subordinates, and improved relationships reduces the fears of employees in regards to sharing their thought and ideas, thus minimizing employee silence. In addition, leaders can encourage and support the employees who need help rather than criticize them for their mistakes. In doing so this will further contribute to the leader–employee social exchange relationship and allow for free exchange of opinions and thoughts, and help in the creation of dialogue.

The results of this study further recommend that leaders who practice the transactional style of leadership, in order to preserve a high level of job satisfaction, apply the transactional leadership behaviors contingent reward, active management-by-exception, and passive management-by-exception more knowingly (this emphasizing the need to train/education the leaders as mentioned previously) so as to limit the negative consequences of each transactional leadership behavior that causes increases in employee silence and to increase the positive impacts in order to reduce employee silence, which will promote employee job satisfaction. As a result, in order to maintain a high level of job satisfaction and to break the silence in the workplace, leaders can improve their use of contingent rewards. A key element of transactional leadership is contingent rewards. By providing contingent rewards that are not seen as used for controlling purposes (for instance, monetary rewards) the transactional leader can be more effective, for instance, with the use of praise.

### 6.3. Limitations and Directions for Further Research

The research does have some limitations that need to be mentioned. Firstly, the variable employee silence was taken as a single construct; however, it is suggested that the sub-dimensions of employee silence (acquiescent silence, defensive silence, and prosocial silence) be studied in order to enable a more in-depth investigation so as to identify the type of silence that is more influential in relationship studies. This would enable leaders to become more aware of the reasons leading to employee silence.

Second, transactional leadership is a style that has been examined in relationships with transformational leadership. Though this study did not do so, it is recommended that future studies examine the study relationship jointly with both transformational and transactional styles of leadership so as to better understand the complementary nature of both styles, as indicated by Bass [33].

Third, this study employed a cross-sectional design. To provide more inclusive results, it is advised that future studies use a longitudinal designs and to combine the questionnaire method of data collection with interviews for a more in-depth understanding of the relationships studies.

Fourth, a convenience sampling method was used; however, it is advised that future studies use a random sampling method to enable a more statistically balanced selection of the population. Additionally, the use of random sampling will allow for more generalizability.

Fifth, respondents were from one public organizational setting. A direction for further research may be to expand the sample and replicate the study at more than one organization.

Finally, this study did not take into account the demographic profile of the respondents. Therefore, a direction for future studies may be to investigate the relevance of age and tenure, for instance, in regards to the relationships studied.

### 7. Conclusions

This study aimed to analyze the relationship between transaction leadership and job satisfaction in a public organizational setting in Jordon through the mediation of employee silence. The theoretical framework was based on the self-determination theory (SDT). The study's main findings revealed that in the public organizational setting, the positive impact of transaction leadership on job satisfaction is suppressed by employee silence, because the transaction leadership behaviors, active and passive management-by-exception, but not contingent reward behavior, have a significant positive impact on employee silence. It is important that public organization leaders become more mindful towards the employee silence phenomenon and recognize how detrimental it can be in the transactional leadership–job satisfaction relationship and therefore take appropriate action to minimize employee silence in the workplace. Organizational leaders also need to be aware of the negative as well as the positive influences of the transactional leadership behaviors (contingent rewards behavior, active management-by-exception, and passive management-by-exception) on employee job satisfaction.

This study, therefore, highlights the important role of the transactional leadership style in ensuring job satisfaction among public organizational employees, a generally under researched sector.

**Author Contributions:** Conceptualization, Y.A.A. and S.E.; Data curation, Y.A.A.; Formal analysis, Y.A.A.; Investigation, Y.A.A. and S.E.; Methodology, Y.A.A. and S.E.; Project administration, Y.A.A. and S.E.; Resources, Y.A.A. and S.E.; Software, Y.A.A.; Supervision, S.E.; Validation, Y.A.A. and S.E.; Visualization, Y.A.A.; Writing—original draft, Y.A.A. and S.E.; Writing—review & editing, Y.A.A. and S.E. All authors have read and agreed to the published version of the manuscript.

**Funding:** There was no external support for this study.

**Institutional Review Board Statement:** A complete set of research ethics principles were followed in this article. The University's Scientific Research Ethics Committee (NEU/SB/2021/947) granted approval for the study's ethical conduct. The participants were also made aware that participation was entirely voluntary and that they might revoke consent at any time. The questionnaire was thus only filled out by individuals who were eager to participate.

**Informed Consent Statement:** Not applicable.

**Data Availability Statement:** On reasonable request, the author will provide the dataset used in this study.

**Acknowledgments:** This article is based on the doctorate dissertation of the first author. This study's co-author is the dissertation supervisor.

**Conflicts of Interest:** The authors state that they have no conflicts of interest.

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
