# Peer review of "Is Silence Golden? The Influence of Employee Silence on the Transactional Leadership and Job Satisfaction Relationship"

_sustainability, doi:10.3390/su142215205_

Round 1

Reviewer 1 Report

I enjoyed reading this manuscript. The manuscript examined this study measured the impact of transactional leadership on job satisfaction as well as the part of employee silence in the leadership-satisfaction relationship in a public organizational setting. The study sample consisted of employees working at the Ministry of Justice in Jordon. The data was tested through confirmatory factor analysis, and correlation and regression analyses were conducted. Results show a strong positive relationship between transactional leadership and job satisfaction, though employees silence as a mediator indicated reduced job satisfaction. Overall, the manuscript is well-written and organized. The rationale of the study is adequately explained. The author(s) have used appropriate and recent sources to develop literature review. The methods and results are adequately discussed. Theoretical and practical implications are well established. I have some minor comments that may benefit the manuscript:

The manuscript should be edited by English native's speakers

The discussion section can be improved by linking the study findings with prior research.

Future research should be discussed in more details

Author Response

Dear Editors and Reviewers

Please find below each reviewer’s comments and our responses. Each comment has been addressed individually.  The requested corrections are indicated with YELLOW highlighting in the manuscript.  An English proofread has also been conducted by a native English-speaker.

In addition, the references have been reorganized as we noticed some errors and within the text the abbreviation for “transactional leadership” (TS) has been replaced with the full written version.

Thank you

REVIEWER 1:

Comment 1: The manuscript should be edited by English native's speakers.

Authors’ Response: The manuscript has undergone an English proofread by a native English speaker.

Comment 2: The discussion section can be improved by linking the study findings with prior research.

Authors’ Response: The discussion has been rewritten to include link the study findings to prior research.

Comment 3: Future research should be discussed in more details.

Authors’ Response: More detail has been provided for the section “Limitations and Directions for Further Research”.

Reviewer 2 Report

The article is interesting, and the researched problem has scientific potential. However, some problems need to be solved:

1. In the abstract, the authors must briefly describe the methods used (the authors forgot to mention SEM).

2. The authors must briefly present the steps of the research (possibly in a figure).

3. The discussion section should be built in the context of dialogue with researchers in the literature review.

4. In my opinion, a solid section of conclusionswill better synthesize the results obtained.

The article has scientific value and can be published after adressing the reported issues.

Author Response

Dear Editors and Reviewers

Please find below each reviewer’s comments and our responses. Each comment has been addressed individually.  The requested corrections are indicated with YELLOW highlighting in the manuscript.  An English proofread has also been conducted by a native English-speaker.

In addition, the references have been reorganized as we noticed some errors and within the text the abbreviation for “transactional leadership” (TS) has been replaced with the full written version.

Thank you

REVIEWER 2:

Comment 1: In the abstract, the authors must briefly describe the methods used (the authors forgot to mention SEM).

Authors’ ResponseAll methods of analysis used in the study have now been included in the abstract.

Comment 2: The authors must briefly present the steps of the research (possibly in a figure).

Authors’ Response: Unfortunately, the authors did not understand exactly what has been requested therefore corrections for this comment have not been provided.

Comment 3: The discussion section should be built in the context of dialogue with researchers in the literature review.

Authors’ Response: The discussion has been rewritten to include linking the study findings to the literature.

Comment 4: In my opinion, a solid section of conclusions will better synthesize the results obtained.

Authors’ Response The conclusion has been rewritten to more clearly synthesize and pinpoint the major findings of the study.

Reviewer 3 Report

1.        Organizational performance is not the variable explored in this study, and the authors are advised to avoid describing variables not relevant to this research topic in the first paragraph of the Introduction section.

2.        Likewise, employee voice is not a variable discussed in this study, and authors are advised to avoid describing variables not relevant to this research topic in the first paragraph of the Practical Implications section.

3.        Practical implications should be improved. The authors should add more implications for managerial practice. These discussions could be more specific and thoughtful (how to do). In other words, the authors should give some practical examples.

4.       I deeply regret to communicate this to the authors and with the hope that my comments and feedback will help them in improving the merit of this paper.

Author Response

Dear Editors and Reviewers

Please find below each reviewer’s comments and our responses. Each comment has been addressed individually.  The requested corrections are indicated with YELLOW highlighting in the manuscript.  An English proofread has also been conducted by a native English-speaker.

In addition, the references have been reorganized as we noticed some errors and within the text the abbreviation for “transactional leadership” (TS) has been replaced with the full written version.

Thank you

REVIEWER 3:

Comment 1: Organizational performance is not the variable explored in this study, and the authors are advised to avoid describing variables not relevant to this research topic in the first paragraph of the Introduction section.

Authors’ ResponseOrganizational performance is not a variable explored in the study however the authors feel the need to emphasize that job satisfaction (one of the study variables) is an important contributor to sustainable organizational performance which is a major concern for all organizations, whether private or public. The authors believe that emphasizing this is also an indicator for the need to study job satisfaction and provides justification for this study.

Comment 2: Likewise, employee voice is not a variable discussed in this study, and authors are advised to avoid describing variables not relevant to this research topic in the first paragraph of the Practical Implications section.

Authors’ Response: The term “employee voice” has been replaced with “employee silence”.

Comment 3:        Practical implications should be improved. The authors should add more implications for managerial practice. These discussions could be more specific and thoughtful (how to do). In other words, the authors should give some practical examples.

Authors’ Response:  The practical implication section has been rewritten to more clearly present, with suggestions/ examples, the implications for managerial practice.